## [Peer Review File · Nature Communications]

Reviewers' comments:

Reviewer #1 (Remarks to the Author):

The authors report atomic-scale shot-noise measurements with a scanning tunneling microscope working in MHz frequency range and find noisy defects in high-T_c superconductor Bi₂212. While the experimental results obtained from their new technique are interesting, I think that some points in their analysis and arguments need to be clarified. The points are listed below followed by some minor comments.

1. Although the authors mention "Mott" in the title as well as in the introduction, I don't find what is related to Mott physics. It would be helpful to clarify new characteristics in the observations distinct from conventional behaviors of dopants in semiconductors.
2. The absence of signature in the topograph does not necessary mean the absence of noisy defects in the topmost BiO layer. It just means the absence of a mechanism causing imaging contrast. If the authors' argument is right, everything in the topmost BiO layer must be visible. But oxygen atoms are not.
3. A natural interpretation of Fig. 2a is that the noisy defects at the center of four Bi atoms, namely Sr site. Therefore, for me, "a natural explanation" (p. 6) is not oxygen dopant atoms but a defect at Sr site, for example, the Bi-Sr antisite reported in ref. 25. Showing the location of the Bi-Sr antisite like Fig. S1 would be convincing to exclude experimentally this possibility. At the same time, Fig. 2a is confusing if the authors intend to say something about the interstitial oxygen dopants. Analysis without averaging would be necessary to resolve interstitial sites.
4. Fig. 2b seems to show the supermodulation as well as the atomic corrugations. Does the noisy defect reside at the bottom of the supermodulation?
5. It is not straightforward to me that discussing TIBB in a superconductor (metal) like in semiconductor. The screening length is generally very short in a metal. Please discuss more why the argument of TIBB in semiconductors where the screening is poor is applicable to a superconductor.

Minor comments:

6. The expression of "the same atomically sized regions" (p. 4) is confusing. Although "atomic" indicates something really atomic in other places, here it means a field of view of 55 nm square.
7. Do "for increasing current" (p. 5) and "for $I > 100$ pA" (p. 5 and the caption of Fig. 3) mean setup current and I_{setup} , respectively?
8. "A" in " A^2eI^2 " is undefined (h and the caption of Fig. 3)
9. One of the authors is not included in the authors list of Supplementary Information.

Reviewer #2 (Remarks to the Author):

The manuscript reports the discovery of new defects in Bi₂Sr₂CaCu₂O_{8+x} using shot-noise STM measurements. These defects, identified as oxygen dopants which eluded previous studies, exhibit an order of magnitude larger Fano factor, and a setup condition dependent dI/dV spectral signature. These defects could provide a new pathway to investigate local charging in high-T_c superconductors. The technique used here is rather novel and the results are very exciting, which in principle warrants publication in a high visibility journal. This paper is likely to motivate other STM groups to pursue similar shot-noise measurements on other materials. But before I can recommend publication, I would require the authors to address the following issues.

- 1) Identifying the chemical origin of noisy defects would be important for future studies. The authors present some evidence that these could be oxygen dopants weakly coupled to the CuO₂ plane. Although this is certainly a possibility, it is hard to prove conclusively. Specifically, it is somewhat difficult to believe that a subset of -1 eV dopants show up as noisy defects due to small

local perturbations that decouple them from the CuO₂ plane, even though their xy position is the same as non-noisy defects. At the very least, I think that the authors should clearly state that noisy defects could in principle also come from other impurities that are not oxygens, or oxygens in a different layer.

In addition, the authors' identification of the the noisy defects as oxygen dopants could be bolstered by the following analysis:

- a) Authors mention that they studied underdoped and optimally-doped samples, so it would be useful to see how the concentration of these defects scales with doping. If these are indeed oxygen dopants, one would expect their density to decrease with decreased hole density.
- b) Is there any correlation between the noisy defects and the structural supermodulation and/or other defects that could point towards local strain changing the defect environment, thus making them less coupled to the CuO₂ plane as the authors hypothesize?

2) The authors state that the oxygen dopants imaged at -1.5 eV are located between 2 Bi atoms. This is in contrast to what has been shown in Ref. 23, where these features are seen directly below the Bi site. I would ask the authors to comment on this discrepancy. The authors also report that they resolved a previously predicted anisotropic shape for defects found at -1.5 eV, but I cannot clearly see this by quickly looking at Ref. 23. Can the authors provide a more clear comparison in the SI between these two studies?

3) The authors note that "As the images in Fig. 1 show, the 'noisy' defects are not located in the topmost BiO layer - there is no signature in topography indicating their presence". I do not believe this is a valid argument. STM measures local density of states, and there are many reasons why a feature would or would not show up in the STM topograph, regardless if it is located in the topmost layer or not.

Minor issues:

Caption Fig. 2: 2d->1d
CuO -> CuO₂ plane

Reviewer #3 (Remarks to the Author):

The manuscript by F. Massee et al, entitled "Noisy defects in a doped Mott insulator", reports on new findings from detailed STM studies on Bi-2212. They reproduce a number of previously found tunneling characteristics ascribed to oxygen defects, and use a new setup to find fingerprints of local (atomic scale) charging effects as measured by a local variation in the tunneling current shot-noise. The main new finding is the identification of local sites with strongly enhanced (non Poissonian) current noise, and these noisy centers are attributed to new (missing) oxygen dopant defect sites weakly coupled to the conducting layers. The latter is based on the location of the noisy defects and their spectroscopic characteristics.

The authors report two different kinds of noisy defects, 2a and 2b, which are located at the same xy position but cause qualitatively different spatial noise patterns. Based on the current-dependence of the dI/dV, it is suggested that different local z positions cause this behavior.

I cannot judge the technical aspects of the AC circuitry and details of the experimental setup for extraction of the shot-noise. For this an experimental STM referee is crucial. However, I can estimate the importance and impact of the findings more generally.

Given the overwhelming literature on STM studies in Bi-2212 it is clearly hard to make important new contributions to this field. However, I believe that the current study does. It is important to thoroughly understand how the Mott insulators turns metallic, and therefore the current work

naturally follows the line of similar STM studies detecting the oxygen defects and their spectroscopy. I find the results trustworthy, the analysis careful, and the discussion as clear as one can be without theoretical modelling backing up the proposals for the various measured features. I recommend publication of this work after the authors have considered the comments/questions below.

1) I missed an estimate of the concentration of the new defects. They are proposed to be the missing ones from previous studies focusing on the -1 and -1,5 eV markers, but is the number now complete, or are there still missing oxygens out there?

2) The authors might consider citing A. Kreisel et al, PRL 114, 217002 (2015) which is the most comprehensive theoretical study I know of, which properly includes Wannier functions in the understanding of STM tunneling on Bi-2212.

3) The authors suggest to "dynamically control doping at the atomic scale", but this is stated just in last sentences of abstract and conclusions without further elaboration. I think a few more sentences would be useful for clarification on what exactly is suggested here.

Reviewers' comments:

Reviewer #1 (Remarks to the Author):

The authors report atomic-scale shot-noise measurements with a scanning tunneling microscope working in MHz frequency range and find noisy defects in high-Tc superconductor Bi2212. While the experimental results obtained from their new technique are interesting, I think that some points in their analysis and arguments need to be clarified. The points are listed below followed by some minor comments.

1. Although the authors mention “Mott” in the title as well as in the introduction, I don’t find what is related to Mott physics. It would be helpful to clarify new characteristics in the observations distinct from conventional behaviors of dopants in semiconductors.

We agree with the referee that the relevance of Mott-ness of the insulating parent compound is not addressed in the manuscript and that mentioning it a number of times may be confusing. Our reason to study optimally doped Bi2212 is that even though it is a high temperature superconductor, one can still consider it as a doped insulator, in this particular case a doped Mott insulator (although if it had been a doped band insulator all the physics we discuss would have applied equally well). Therefore, we expected semi-conductor physics like charging and de-charging of dopants still to be possible/present, as we indeed find. Even though these processes are on a much shorter time scale due to the more metallic (but not truly metallic) nature of the system, all the processes well-known from semi-conductor research, including tip induced band bending, apply equally well to our system. To avoid confusion we have de-emphasised the Mott-ness of the insulating state in the main text (abstract and first paragraph) and changed the title.

2. The absence of signature in the topograph does not necessary mean the absence of noisy defects in the topmost BiO layer. It just means the absence of a mechanism causing imaging contrast. If the authors’ argument is right, everything in the topmost BiO layer must be visible. But oxygen atoms are not.

The referee is correct. We meant to argue that there are no distinct signatures of the defect in the low energy topography: all Bi atoms are present and none are unusually distorted. This makes it unlikely – but indeed not impossible – for a foreign object to be present in the top-most layer at the noisy locations. We have changed the text accordingly (page 4).

3. A natural interpretation of Fig. 2a is that the noisy defects at the center of four Bi atoms, namely Sr site. Therefore, for me, “a natural explanation” (p. 6) is not oxygen dopant atoms but a defect at Sr site, for example, the Bi-Sr antisite reported in ref. 25. Showing the location of the Bi-Sr antisite like Fig. S1 would be convincing to exclude experimentally this possibility. At the same time, Fig. 2a is confusing if the authors intend to say something about the interstitial oxygen dopants. Analysis without averaging would be necessary to resolve interstitial sites.

It is true that there is a Sr atom directly below the BiO layer, positioned in between a square of four Bi atoms, and that a substitution on this site can give rise to resonances as shown in Ref 25. However, the Bi-Sr substitution the referee refers to shows up at positive energies, not the negative energies we observe our defects – a resonance at negative energy in Ref. 25 occurs when Pb replaces Bi, which is not possible in our Pb-free system. We have added this discussion to the manuscript (page 7).

We agree that analysis without averaging, similar to Ref. 25, would be ideal. At the same time, theoretical studies such as Ref. 19 show that there are only a limited number of energetically

favourable interstitial sites available to the oxygen dopants. The fact that our averaged topography shows structure supports this idea and gives us the location of the interstitial site where the noisy defects reside to a good approximation. We have added this information to the manuscript (page 4).

Analysis without averaging, which relies on intricate algorithms to correct for the intrinsically distorted nature of the material, will provide a measure of the spread around this average site, which may indeed be relevant for finding out why some dopants are noisy while most are not. However, given the apparent size of the defects in the noise and dI/dV , and the distorted nature of the material, it is not a priori evident that the error in determining this spread is smaller than the spread itself. As we do not believe the spread is essential at this point to support our main findings, we opted to refrain from such intricate analysis.

4. Fig. 2b seems to show the supermodulation as well as the atomic corrugations. Does the noisy defect reside at the bottom of the supermodulation?

This is indeed an interesting observation. However, on other samples the super-modulation does not appear in the averaged topography. We have added discussion to the main text (page 8) and a figure to the Supplementary Information (new section 2).

5. It is not straightforward to me that discussing TIBB in a superconductor (metal) like in semiconductor. The screening length is generally very short in a metal. Please discuss more why the argument of TIBB in semiconductors where the screening is poor is applicable to a superconductor.

This is a crucial point. From c-axis resistivity measurements it is known that the system is not metallic perpendicular to the CuO_2 planes, i.e. whereas the CuO_2 plane is superconducting with short in-plane screening lengths, the out of plane layers are insulating/semiconducting. The observation of charge dynamics in the system confirms this. At the energies where we perform our measurements, we are essentially probing the semiconductor physics of the out of plane layers, with poor screening by the CuO_2 states. We have added a discussion to clarify this issue (page 8).

Minor comments:

6. The expression of “the same atomically sized regions” (p. 4) is confusing. Although “atomic” indicates something really atomic in other places, here it means a field of view of 55 nm square. In this particular instance we actually mean the regions of atomic scale dimensions within the 55nm square field of view (i.e. the dark spots in Fig. 1d), not the entire 55nm field of view. We have rephrased the sentence to remove the confusion (page 4).

7. Do “for increasing current” (p. 5) and “for $I > 100$ pA” (p. 5 and the caption of Fig. 3) mean setup current and I_{setup} , respectively?

We apologise for the confusion arising from mixing current (I), and setup current (I_{setup}) in the text. We have changed the main text (page 5), as well as the caption of Fig. 3. We have further added to the legend of figures 3g,h that the numbers listed are the setup current.

8. “A” in “ A^2eI^2 ” is undefined (h and the caption of Fig. 3)

We have added the value of A to the caption of Fig. 3.

9. One of the authors is not included in the authors list of Supplementary Information.

Our apologies, the author has been added.

Reviewer #2 (Remarks to the Author):

The manuscript reports the discovery of new defects in $\text{Bi}_2\text{Sr}_2\text{CaCu}_2\text{O}_{8+x}$ using shot-noise STM measurements. These defects, identified as oxygen dopants which eluded previous studies, exhibit an order of magnitude larger Fano factor, and a setup condition dependent dI/dV spectral signature. These defects could provide a new pathway to investigate local charging in high- T_c superconductors. The technique used here is rather novel and the results are very exciting, which in principle warrants publication in a high visibility journal. This paper is likely to motivate other STM groups to pursue similar shot-noise measurements on other materials. But before I can recommend publication, I would require the authors to address the following issues.

1) Identifying the chemical origin of noisy defects would be important for future studies. The authors present some evidence that these could be oxygen dopants weakly coupled to the CuO_2 plane. Although this is certainly a possibility, it is hard to prove conclusively. Specifically, it is somewhat difficult to believe that a subset of -1 eV dopants show up as noisy defects due to small local perturbations that decouple them from the CuO_2 plane, even though their xy position is the same as non-noisy defects. At the very least, I think that the authors should clearly state that noisy defects could in principle also come from other impurities that are not oxygens, or oxygens in a different layer.

In part we refer back to our answer to a similar comment by referee 1. Additionally, we stress that not all conventional dopants are exactly at $E = -1$ eV: they can be found within a range of roughly ± 100 mV around 0.9 eV. This is important, because it means that even the conventional dopants can thus occupy a range of environments. We argue here that some environments are in a sense even more extreme leading to the dopants with charge dynamics, which gives both previously most likely seen, but ignored, features in the density of states and excess current noise. This is further supported by the fact that most noisy defects appear at lower absolute energies than the conventional ones, indicating they are more weakly coupled to the CuO_2 plane / closer to the BiO surface. We have stressed these points now in the main text (page 6 and 8). Despite the evidence that we have, we indeed cannot fully exclude other impurities as the cause for the excess noise and concomitant features in the density of states. We have added this information to the main text (page 7).

In addition, the authors' identification of the noisy defects as oxygen dopants could be bolstered by the following analysis:

a) Authors mention that they studied underdoped and optimally-doped samples, so it would be useful to see how the concentration of these defects scales with doping. If these are indeed oxygen dopants, one would expect their density to decrease with decreased hole density.

This is an excellent suggestion by the referee and is indeed one of the reasons we studied materials with a different doping concentration. Unfortunately the number of noisy defects is relatively low (see also our comment to the first comment of referee 3) – too low for a statistically relevant analysis of the doping dependence. We added this discussion to the main text (footnote page 7).

b) Is there any correlation between the noisy defects and the structural supermodulation and/or other defects that could point towards local strain changing the defect environment, thus making them less coupled to the CuO_2 plane as the authors hypothesize?

The referee raises an interesting point. Although the super-modulation remains visible in the averaged topography in Fig. 2b, this is not the case for every sample (see also our reply to comment 4 of referee 1), making a clear-cut link unlikely. To find out if there is a correlation between the noisy defects and the conventional ones we refer to Fig. S1c. Noisy defects can be

seen both in areas with a relatively large number of regular dopants, but also in areas void of them. Possibly the defects in latter regions are then predominantly located near the crest of the super-modulation, but the limited statistics makes this type of analysis rather speculative. We have added a discussion to the main text (page 8).

2) The authors state that the oxygen dopants imaged at -1.5 eV are located between 2 Bi atoms. This is in contrast to what has been shown in Ref. 23, where these features are seen directly below the Bi site. I would ask the authors to comment on this discrepancy. The authors also report that they resolved a previously predicted anisotropic shape for defects found at -1.5 eV, but I cannot clearly see this by quickly looking at Ref. 23. Can the authors provide a more clear comparison in the SI between these two studies?

Fig. 2e of Ref. 23 indeed shows the main intensity of the experimental differential conductance to be closer towards one of two Bi site – although not exactly on the Bi site. The simulation in Ref. 23 on the other hand (Fig. 2h) assumes the oxygen site to be in between two Bi atoms, resulting in a dumbbell shaped signature in the dI/dV . The main intensity in dI/dV for this simulation falls on the two Bi atoms neighbouring the oxygen site. Whereas this dumbbell shape is not seen experimentally in Ref. 23 – perhaps only one half of the dumbbell is detected – we observe the entire dumbbell. We do not know why the data in Ref. 23 shows only part of the dumbbell, but based on the energy of the feature we are looking at the same object. The agreement between the theory in Ref. 23 and our dI/dV measurement strongly support the site of the oxygen dopant to be in between the Bi atoms as assigned by Ref. 23. We have added discussion to the SI (section 1).

3) The authors note that "As the images in Fig. 1 show, the ‘noisy’ defects are not located in the topmost BiO layer - there is no signature in topography indicating their presence". I do not believe this is a valid argument. STM measures local density of states, and there are many reasons why a feature would or would not show up in the STM topograph, regardless if it is located in the topmost layer or not.

We refer back to our answer to comment 2 of referee 1. We have adjusted this statement in the main text (page 4).

Minor issues:

Caption Fig. 2: 2d->1d

CuO -> CuO₂ plane

We thank the referee for his careful reading and have corrected these errors.

Reviewer #3 (Remarks to the Author):

The manuscript by F. Massee et al, entitled “Noisy defects in a doped Mott insulator”, reports on new findings from detailed STM studies on Bi-2212. They reproduce a number of previously found tunneling characteristics ascribed to oxygen defects, and use a new setup to find fingerprints of local (atomic scale) charging effects as measured by a local variation in the tunneling current shot-noise. The main new finding is the identification of local sites with strongly enhanced (non Poissonian) current noise, and these noisy centers are attributed to new (missing) oxygen dopant defect sites weakly coupled to the conducting layers. The latter is based on the location of the noisy defects and their spectroscopic characteristics.

The authors report two different kinds of noisy defects, 2a and 2b, which are located at the same xy position but cause qualitatively different spatial noise patterns. Based on the current-dependence of the dI/dV , it is suggested that different local z positions cause this behavior.

I cannot judge the technical aspects of the AC circuitry and details of the experimental setup for extraction of the shot-noise. For this an experimental STM referee is crucial. However, I can estimate the importance and impact of the findings more generally.

Given the overwhelming literature on STM studies in Bi-2212 it is clearly hard to make important new contributions to this field. However, I believe that the current study does. It is important to thoroughly understand how the Mott insulators turns metallic, and therefore the current work naturally follows the line of similar STM studies detecting the oxygen defects and their spectroscopy. I find the results trustworthy, the analysis careful, and the discussion as clear as one can be without theoretical modelling backing up the proposals for the various measured features. I recommend publication of this work after the authors have considered the comments/questions below.

1) I missed an estimate of the concentration of the new defects. They are proposed to be the missing ones from previous studies focusing on the -1 and -1,5 eV markers, but is the number now complete, or are there still missing oxygens out there?

As can be seen from Figure 1d, there are on the order of 50 noisy defects in a 55nmx55nm field of view. If we consider only the topmost layers (CuO₂, SrO, BiO), a single unit cell of 0.54nmx0.54nm contains 2 Cu atoms. Our noisy defect density of roughly 50 on 10⁴ unit cells then gives a doping $x \sim 0.005$, which is rather small compared to the nominal doping concentration of $x \sim 0.16$. Hence, although the defects we find add to the number of ‘known’ defects, their number is still short of the total expected number: there are still missing oxygens out there. We have added a note to the main text with this information (footnote page 7).

2) The authors might consider citing A. Kreisel et al, PRL 114, 217002 (2015) which is the most comprehensive theoretical study I know of, which properly includes Wannier functions in the understanding of STM tunneling on Bi-2212.

We thank the referee for his suggestion and have added the reference (page 2).

3) The authors suggest to “dynamically control doping at the atomic scale”, but this is stated just in last sentences of abstract and conclusions without further elaboration. I think a few more sentences would be useful for clarification on what exactly is suggested here.

We thank the referee for his suggestion, we have elaborated on this statement in the conclusion.

Reviewers' comments:

Reviewer #1 (Remarks to the Author):

The authors addressed most of the points I raised previously. Although some of the replies are more modest and less conclusive than I had expected, I would like to respect the authors' arguments, which would be decent for publication, except the argument about the screening length. As the authors mentioned in their reply, the temperature dependence of c-axis resistivity is semiconducting. But it's at elevated temperatures above T_c , whereas the authors measurements were done in the superconducting state at 1.8 K. The point to be discussed here is the TIBB in the superconducting state. Although the superconducting properties also show two-dimensional characters, the sample is a superconductor, and not a semiconductor. The additional description in the amended text "this may not be true for Bi2212 due to its strongly two dimensional character" is inadequate. I would like the authors to clarify why the argument about the TIBB is applicable to the superconducting state even though "the screening length is typically too short for such TIBB to play any role".

Reviewer #2 (Remarks to the Author):

The authors have sufficiently addressed most of my concerns. I also read through the comments by other referees and I believe the authors have sufficiently addressed those as well. I support the publication of the paper in Nature Communications.

Reviewer #3 (Remarks to the Author):

The authors have answered my requests and questions, and modified the manuscript accordingly. I recommend publication.

Reply to the referee comments of the re-submission:

We would like to thank all reviewers once again for their time and are pleased that they agree to all but one of the changes we have made following their reviews. The only remaining issue concerns the justification of our use of semiconductor physics (tip induced band bending) in a superconductor.

Referee #1 points out that the semi-conducting c-axis transport that we use in our argument is a high temperature characteristic while we measure at $T \sim 0$ K. We would like to point out that measurements on cuprates where superconductivity is suppressed by a magnetic field show semiconducting c-axis transport down to the lowest temperature (see e.g. *PRL* **77**, 2065 (1997)). With this in mind, the referee further states that we are in the superconducting state, not the normal state. Here we stress that the screening length in a superconductor is governed by the same Thomas-Fermi length-scale as in the normal state (see *PRB* **70**, 226503 (2004)). If anything the screening length in the superconducting state will be longer than that in the normal state (see also *PRB* **70**, 226504 (2004)). A recent experiment using conventional superconductors (*Nature Nanotechnology* **13**, 802-805 (2018)) indeed shows this to be the case. It is therefore highly plausible that the screening length along the c-axis in Bi2212 is sufficiently long to allow for tip induced band bending to occur.

We have amended the text and added two references to convey this message more clearly, the new text and references are highlighted in red.

REVIEWERS' COMMENTS:

Reviewer #1 (Remarks to the Author):

The authors have replied my final question and clarified their discussion in the main text. I recommend publication.